# Electrochemical Degradation of Sulfamethoxazole Enhanced by Bio-Inspired Iron-Nickel Encapsulated Biochar Particle Electrode

**DOI:** 10.3390/ijms252413579

**Published:** 2024-12-19

**Authors:** Shuang Geng, Jingang Yao, Lei Wang, Yangyang Wang, Xiaoshu Wang, Junmin Li

**Affiliations:** 1School of Agricultural Engineering and Food Science, Shandong University of Technology, Zibo 255000, China; 22503040006@stumail.sdut.edu.cn (S.G.); 22503040002@stumail.sdut.edu.cn (J.L.); 2School of Materials and Environmental Engineering, Institute of Urban Ecology and Environment Technology, Shenzhen Polytechnic University, Shenzhen 518055, China; 22503040007@stumail.sdut.edu.cn (Y.W.); xshuwang@szpu.edu.cn (X.W.); 3School of Resource and Environmental Engineering, Wuhan University of Technology, Wuhan 430070, China

**Keywords:** bio-inspired catalyst, FeNi@BC, sulfamethoxazole, particle electrode, electrocatalysis

## Abstract

In the electrocatalytic (EC) degradation process, challenges such as inefficient mass transfer, suboptimal mineralization rates, and limited current efficiency have restricted its broader application. To overcome these obstacles, this study synthesized spherical particle electrodes (FeNi@BC) with superior electrocatalytic performance using a bio-inspired preparation method. A three-dimensional electrocatalytic oxidation system based on FeNi@BC electrode, EC/FeNi@BC, showed excellent degradation efficiency of sulfamethoxazole (SMX), reaching 0.0456 min^−1^. Quenching experiments and electron paramagnetic resonance experiments showed that the excellent SMX degradation efficiency in the EC/FeNi@BC system was attributed to the synergistic effect of multiple reactive oxygen species (ROS) and revealed their evolution path. Characterization results showed that FeNi_3_ generated in the FeNi@BC electrode was a key bimetallic active site for improving electrocatalytic activity and repolarization ability. More importantly, the degradation pathway and reaction mechanism of SMX in the EC/FeNi@BC system were proposed. In addition, the influencing factors of the reaction system (voltage, pH, initial SMX concentration, electrode dosage, and sodium sulfate concentration, etc.) and the stability of the catalyst (maintained more than 81% after 5 cycles) were systematically evaluated. This study may provide help for the construction of environmentally friendly catalytic and efficient degradation of organic pollutants.

## 1. Introduction

Advanced oxidation technologies are highly regarded in the field of water research [1,2]. Among these, electrocatalytic oxidation has emerged as a particularly effective and environmentally sustainable approach. It offers several advantages, including high efficiency, minimal secondary pollution, low operational costs, and the use of simple equipment [3,4]. Despite these benefits, the technology faces several challenges. Bu et al. identified difficulties in detecting free radicals during the degradation process and emphasized the need for further research into the roles of other reactive oxygen species (ROS), such as •O_2_^−^, •OH, ^1^O_2_, and SO_4_^−•^ [5]. Additionally, the degradation mechanisms remain insufficiently explored. Furthermore, conventional two-dimensional electrocatalytic oxidation methods are hindered by issues such as reduced current efficiency, lower mineralization rates, and suboptimal mass transfer [6].

As an advanced oxidation technique, three-dimensional electrocatalytic oxidation offers several advantages, including operation under mild conditions, prevention of secondary pollution, and enhanced mineralization efficiency in wastewater treatment [7,8]. The incorporation of particle electrode materials in this three-dimensional system can significantly boost pollutant degradation performance [9]. For instance, Wan et al. utilized a combination of metals (CuO, Sb_2_O_3_, NiO, TiO_2_, ZnO, and MnO) deposited on honeycomb-structured initiated carbon to strengthen the electrocatalytic degradation of p-aminophenol in wastewater [10]. This approach resulted in a dramatic improvement in removal efficiency, increasing from 50.20% to 100%. This highlights that the three-dimensional electrocatalytic oxidation system is more efficient than its two-dimensional counterpart. Additionally, the combined effects of various metals working synergistically further enhance electrocatalytic performance [10]. However, particle electrode materials are not without limitations. Low-impedance materials such as activated carbon, metal, and graphite can cause short-circuiting. To address this issue, researchers have incorporated materials like plastics, glass beads, and quartz sand to increase impedance. Unfortunately, these materials often differ significantly in density, shape, and size from particle electrodes, causing layering between the electrode and insulating materials, which may lead to short circuits during operation [11,12]. Consequently, substantial research is now focused on developing particle electrode materials that offer superior insulation, enhanced stability, and improved catalytic efficiency.

Iron (Fe) and nickel (Ni) ions are essential nutrients that plants absorb during growth. These ions are uniformly distributed within the biomass and can serve as key raw materials for producing biochar enriched with Fe and Ni [13]. Previous studies have shown that while plants contain over 8.75 wt% Fe and 3.25 wt% Ni, pyrolysis and carbonization increase these concentrations in bio-inspired FeNi5@BC9 biochar to 24.63 wt% Fe and 15.21 wt% Ni [14]. Through a simple pyrolysis process under anaerobic conditions, Fe and Ni from biomass can be encapsulated within the bio-inspired FeNi5@BC9, resulting in a uniform distribution of these elements throughout the material, potentially enhancing its physicochemical properties [15]. When effectively utilized, these biochar catalysts could overcome the limitations of synthetic Fe and Ni biochar (FeNi-BC). Specifically, Fe^2+^ and N^i2+^ may be released during the electrocatalytic process, generating ROS that aid in the degradation of organic pollutants. Moreover, the high performance of bio-inspired FeNi@BC could reduce the need for Fe-rich biomass. However, the chemical transformations of Fe and Ni elements during pyrolysis and their roles in electrocatalytic activation remain insufficiently understood.

In this study, the principle of “using waste to treat waste and transforming waste into valuable resources” was employed. For the first time, FeNi@BC, a catalyst with exceptional structural stability and catalytic performance, was developed. This catalyst was incorporated into an electrochemical reaction chamber to establish a three-dimensional catalytic oxidation system (EC/FeNi@BC), designed to degrade SMX, a model sulfa-antibiotic used to evaluate the system’s efficacy. The involvement of ROS was systematically investigated through free radical scavenging experiments. The electrocatalytic performance of FeNi@BC was assessed using an electrochemical workstation, while X-ray photoelectron spectroscopy (XPS) was employed to identify the active catalytic sites on the particle electrodes. The degradation mechanism of SMX was further explored through LC-MS analysis. Finally, the stability of the catalyst was evaluated through cyclic experiments involving SMX.

## 2. Results and Discussion

### 2.1. Characterization of the Catalysts

The morphologies and microstructures of the samples were analyzed using SEM, as illustrated in Figure 1a–f. The BC9 sample exhibited well-defined porous carbon structures with a smooth surface (Figure 1a). The introduction of Fe and Ni increased the number of surface pores, likely due to chemical reactions within the biochar [16]. Figure 1b–f shows that the addition of irregularly shaped particles resulted in a rough surface, in stark contrast to the smooth surface of the unmodified BC. This indicates that Fe-Ni nanoparticles were successfully incorporated into both FeNi@BC and FeNi-BC catalysts. As the pyrolysis temperature increased, Fe-Ni atoms aggregated to form spherical particles with a uniform morphology, suggesting that high-temperature calcination is a major factor in the formation of Fe-Ni alloys. The images also revealed improved clarity, indicating enhanced conductivity. In contrast, the surface of FeNi-BC appeared irregular (Figure 1f). Further observations revealed that while FeNi-BC primarily exposed Fe-Ni atoms on its surface, FeNi5@BC9 displayed a more uniform distribution of Fe-Ni atoms throughout the framework. This suggests that the carbon coating on the Fe-Ni particles in FeNi5@BC9 protects the active components from loss and oxidation. The distinctive structure of FeNi5@BC9 also contributes to its resistance to bulk oxidation, demonstrating exceptional stability. BET analysis further revealed that the spherical Fe-Ni particles, along with newly formed Fe and Ni oxides, significantly increased the SSA of FeNi5@BC9. The hierarchical FeNi5@BC9, with its high concentration of Fe and Ni, showed a much larger SSA compared to FeNi-BC and BC. For instance, the SSA of FeNi5@BC9 reached 457.78 m^2^ g^−1^, which is approximately 65 times greater than that of BC (6.99 m^2^ g^−1^) and about 1.5 times larger than FeNi-BC (295.78 m^2^ g^−1^) (Appendix A). This increase suggests the formation of additional aromatic carbon substrates [17]. The EDS spectra reveal that the surface composition of FeNi5@BC9 contains 21.0 wt% of Fe and Ni, which is significantly higher than the 10.7 wt% observed in FeNi-BC (Appendix A). Moreover, a distinct trend emerges with increasing pyrolysis temperature, as the metal content follows the order: FeNi5@BC9 (21.0 wt%) > FeNi5@BC7 (9.3 wt%) > FeNi5@BC5 (5.9 wt%).

The chemical states of the Fe and Ni species were analyzed using the XRD patterns of the synthesized catalysts. At 2θ = 26.6°, distinct peaks were observed, corresponding to carbon (PDF#97-002-9123) and reflecting the material’s high degree of crystallinity (Figure 1g). In the FeNi5@BC9 sample, diffraction peaks at 2θ = 35.4° and 44.7° were assigned to the (311) planes of Fe_3_O_4_ (PDF#97-015-9975) and the (110) planes of Fe (Iron, delta, PDF#97-004-4863), respectively. Additionally, peaks at 2θ = 36.5° and 50.6° corresponded to the (301) plane of NiO (PDF#97-018-2948) and (200) plane of FeNi_3_ (PDF#97-063-2930) [18,19,20] (Figure 1h). In contrast, the FeNi-BC sample exhibtied The most prominent peaks primarily corresponding to Fe (Iron, delta) and NiO (Figure 1i). These findings suggest that the bio-inspired encapsulation method effectively promoted bimetallic bonding, as Fe and Ni ions were integrated into plant cells via metabolic processes. Notably, new diffraction peaks at 2θ = 42.3° and 74.0°, corresponding to the (111) and (220) planes, were exclusively observed in FeNi5@BC9. These peaks indicate the formation of austenite (CFe_15.1_, PDF#00-052-0512) as a result of the bio-inspired encapsulation process. Furthermore, the findings imply that Fe_3_O_4_ was effectively reduced to Fe (Iron, delta) and CFe_15.1_ by carbon species during pyrolysis at 900 °C in FeNi5@BC9.

XPS was employed to analyze the atomic valence and chemical bonding in both FeNi5@BC9 and FeNi-BC catalysts. The survey spectra confirmed the presence of Fe, Ni, C, N, and O in both samples. The high-resolution C 1s spectra, shown in Appendix A, were deconvoluted into four distinct peaks at 287.7 eV, 286.8 eV, 285.6 eV, and 284.8 eV, corresponding to the O–C=O, C–N, C–P, and C–C/C=C species, respectively [21]. This indicates the formation of carbon doped with O, N, and P. In the high-resolution Fe spectrum of FeNi5@BC9, peaks at 711.8 eV and 725.3 eV were assigned to the Fe 2p_3/2_ and Fe 2p_1/2_ states of Fe^3+^, respectively. Additionally, the peak at 710.9 eV was associated with the Fe 2p_3/2_ state of Fe^2+^ [22,23], while a satellite peak was observed at 727.9 eV [24]. Figure 1i shows the high-resolution Ni 2p spectrum, where the deconvoluted peaks at 874.8 eV and 855.5 eV correspond to Ni 2p_1/2_ and Ni 2p_3/2_ binding energies of Ni^2+^, with additional peaks representing satellite signals [25,26]. Notably, in FeNi5@BC9, the Fe 2p_3/2_ peak for Fe^3+^ at 711.8 eV and the Ni 2p_3/2_ peak for Ni^2+^ at 855.5 eV were shifted positively by 0.6 eV and 0.1 eV, respectively, compared to those in FeNi-BC. These shifts suggest that the bio-inspired preparation method effectively modified the electron cloud densities around the Ni and Fe sites, optimizing their electronic structures [27]. Additionally, in the biochar prepared using this bio-inspired method (FeNi5@BC9), the binding energies of Fe and Ni sites exhibited varied shifts, indicating changes in the electron cloud density at the active centers. Prior research has shown that this approach adjusts the electronic configuration of the carbon substrate, thereby enhancing the interaction between the support and the functional material [23,28]. This alteration in the electronic configuration of both the substrate and the active sites markedly improved the electrocatalytic degradation of SMX. As shown in the O 1s spectra (Figure 1l), both materials exhibit characteristic peaks at binding energies of 532.2 eV, 533.7 eV, and 536.4 eV, corresponding to surface-adsorbed oxygen species (O_ads_, O_2_^−^ or O_2_^2−^), lattice oxygen (O_latt_, O_2_^−^), and surface hydroxyl species (O_ads_O-H), respectively [29,30]. Notably, FeNi5@BC9 (73.16%) demonstrates a higher proportion of surface-adsorbed oxygen species compared to FeNi-BC (70.40%). According to existing literature, the concentration of surface-adsorbed reactive oxygen species is closely linked to the density of oxygen vacancies (OVs). A higher density of OVs not only promotes the activation of O_2_, resulting in the formation of reactive oxygen species on the surface, but also facilitates the migration of O_latt_ to the surface, thereby further enhancing the electrocatalytic process. These findings highlight the superior catalytic performance of FeNi5@BC9 compared to FeNi-BC.

### 2.2. Electrochemical Measurement of the Catalysts

To evaluate the electrochemical properties of various Fe-Ni-enriched biochar materials, an electrochemical workstation was used to assess the performance of BC, FeNi-BC, and FeNi5@BC9 particle electrodes. Detailed specifications of the electrochemical workstation are provided in the Appendix A [31].

The cyclic voltammetric (CV) properties of BC, FeNi-BC, and FeNi5@BC9, as shown in Figure 2a, highlight the superior stability of FeNi5@BC9. The CV curves for the BC electrode did not exhibit any redox peaks, and the FeNi-BC electrodes lacked clear reductive features. In contrast, FeNi5@BC9 displayed a significantly larger area within the rectangular frame, indicating an expanded catalytically active surface and a greater number of active sites. A larger area in the cyclic voltammogram typically correlates with higher specific capacity of the material [31,32]. Additionally, the CV data demonstrated that FeNi5@BC9 exhibits excellent repolarization capability within the degradation system.

Figure 2b shows the Tafel polarization curves for BC, FeNi-BC, and FeNi5@BC9. The data indicate that FeNi5@BC9 exhibits a lower corrosion potential (−0.6261 V) compared to BC (−0.2398 V) and FeNi-BC (−0.4749 V). Additionally, FeNi5@BC9 displays significantly higher current densities, likely due to enhanced electron transfer facilitated by the uniform distribution of Fe and Ni. Corrosion current values derived from electrochemical data reveal that FeNi5@BC9 has a corrosion current of 3.172 × 10^−^^5^ A, which is higher than that of FeNi-BC (1.604 × 10^−^^5^A) but lower than BC (4.730 × 10^−^^5^ A). These findings suggest that FeNi5@BC9 possesses superior electron carrier capabilities.

Figure 2c presents the Nyquist plot along with the corresponding fitted parameters derived from electrochemical impedance spectroscopy (EIS). As noted in previous research [33], the diameter of the depressed semicircle in the high-frequency region indicates the resistance to electron transfer, where a smaller arc radius indicates lower electron transport resistance. The larger semicircular diameters observed for BC and FeNi-BC, compared to FeNi5@BC9, suggest that FeNi5@BC9 offers superior electron transport characteristics with reduced resistance. The equivalent circuit simulation reveals that FeNi5@BC9 significantly reduces the charge transfer resistance (Rct) from 42.21 Ω to 26.38 Ω compared to FeNi-BC. This reduction indicates more efficient electron exchange between the electrode and electrolyte, resulting in enhanced electrocatalytic activity [34]. Such improvement is likely attributed to the higher degree of graphitization and semiconductor properties imparted by the bio-inspired preparation method [35]. These findings validate the effectiveness of the bio-inspired preparation method in minimizing the interaction between Fe-Ni and carbon species, enabling sustained electron transfer and significantly enhancing the electrocatalytic performance of FeNi5@BC9.

### 2.3. Catalytic Performance of the Catalysts

To evaluate SMX degradation across various systems, the contribution of adsorption to the removal efficiency was examined. Particle electrode materials, including BC9, FeNi-BC, and FeNi5@BC9, were immersed in a solution containing SMX at a concentration of 20 mg L^−^^1^. As illustrated in Figure 3a, none of the three catalysts exhibited significant adsorption of SMX under the tested conditions (volts = 8 V, pH = 3, the dose of catalysts = 10 mg, Na_2_SO_4_ dose = 0.10 M, and the adsorption time = 120 min). However, the EC, EC/BC9, and EC/FeNi-BC systems achieved SMX degradation efficiencies of 57.1%, 71.03%, and 76.57%, respectively. Notably, the introduction of FeNi5@BC9 into the EC system achieved a 99.49% SMX removal efficiency. The degradation process adhered pseudo-first-order kinetics, with a rate constant of 0.0456 min^−1^. The SMX degradation rate in the EC/FeNi5@BC9 system (0.0456 min^−1^) was observed to be 3.8 times higher than that in the EC/FeNi-BC system (0.0121 min^−1^) and 5.4 times higher compared to the EC system (0.0084 min^−1^), as depicted in Figure 3a. This underscores the substantial enhancement in SMX degradation efficiency when using the EC/FeNi5@BC9 system.

Energy consumption plays a critical role in determining the feasibility of electrochemical applications, with the amount of energy used during the degradation process being closely linked to the applied voltage. In this study, low voltage conditions ranging from 2 to 10 V were employed to prioritize energy conservation and prevent electrode damage due to excessive voltage. To investigate the impact of cell voltage on SMX removal, single-factor experiments were performed using the EC/FeNi5@BC9 three-dimensional particle electrode reactor. The results indicated that voltage significantly influenced SMX removal efficiency. At 2 V, the degradation efficiency after 120 min was only 71.03%. Increasing the voltage to 6 V improved the degradation efficiency to 91.91%, with the highest efficiency observed at 8 V, reaching 95.27%. However, a slight decline in efficiency was observed at 10 V, with the removal efficiency decreasing to 92.77% (Figure 3b). Increasing the cell voltage enhances the power field distribution within the reactor and increases the driving force following the re-polarization of the particle electrode, resulting in improved degradation performance [36]. Within the tested range, the reaction rate constant rose with increasing voltage, and the optimal degradation efficiency was achieved at 8 V, although even at lower voltages, high degradation rates were attained.

The treatment efficiency was significantly influenced by the initial pH of the SMX solution, with higher removal rates observed at lower pH levels. Specifically, degradation efficiencies at pH values of 3, 5, 7, 9, and 11 were 99.49%, 96.45%, 95.27%, 88.77%, and 84.00%, respectively (Figure 3c). This trend is likely due to the increased generation of hydrogen peroxide (H_2_O_2_) under acidic conditions. Under lower pH, dissolved oxygen in the electrolyte gains two electrons at the cathode, producing H_2_O_2_. This H_2_O_2_ then reacts with active metal oxides, such as Fe and Ni, in the FeNi5@BC9 cathode to generate ROS like •OH, •O_2_^−^, and ^1^O_2_ (Equations (2)–(8)) [37,38]. The degradation of SMX proceeds through primary mechanisms: direct decomposition and indirect oxidative degradation [39]. In direct decomposition, SMX is rapidly broken down by oxygen-containing radicals generated at the electrode [40]. Indirect oxidative degradation, on the other hand, involves the transformation of SMX into smaller molecules through H_2_O_2_, which decomposes to form reactive species [41]. This process is slower and necessitates higher concentrations of free radicals [42]. Acidic conditions promote the production of •OH, enhancing the oxidation rate [43]. In contrast, alkaline conditions reduce SMX degradation efficiency due to several factors, including repulsion between the negatively charged SMX molecules and the catalyst surface, inhibition of •OH production, and a decrease in oxygen evolution potential at the anode [44]. Furthermore, under weakly alkaline conditions, SMX molecules exist as anions, increasing their susceptibility to nucleophilic attack by •OH [7,45].
(1)$$O_{2}+2H^{+}+2e^{-}\to H_{2}O_{2}$$


(2)
$$H_{2}O_{2}+{\mathrm{Me}}^{n+}\to {\mathrm{Me}}^{(n+1)+}+∙\mathrm{OH}+{\mathrm{OH}}^{-}(\mathrm{Me}=\mathrm{Fe},\;\mathrm{Ni})$$



(3)
$${\mathrm{Me}}^{(n+1)+}+H_{2}O_{2}\to {\mathrm{Me}}^{n+}+∙{O_{2}}^{-}+2H(\mathrm{Me}=\mathrm{Fe},\;\mathrm{Ni},\;{k^{+}}_{1}=2.0\times 10^{-3}\;M^{-1}\;s^{-1})$$



(4)
$${\mathrm{Me}}^{(n+1)+}+∙{O_{2}}^{-}\to {\mathrm{Me}}^{n+}+O_{2}(\mathrm{Me}=\mathrm{Fe},\;\mathrm{Ni},k_{2}=2.0\times 10^{3\text{-}5}\;M^{-1}\;s^{-1})$$



(5)
$$2∙{O_{2}}^{-}/∙HO_{2}\to H_{2}O_{2}+O_{2}(k_{3}=2.3\times 10^{6}\;M^{-1}\;s^{-1})$$



(6)
$$\mathrm{OV}\;(e^{-})\;+O_{2}\to ∙{O_{2}}^{-}\;(k_{4}=2.0\times 10^{10}\;M^{-1}\;s^{-1})$$



(7)
2∙O2−+2H2O →H2O2+2OH−+O21


As the initial concentration of SMX increased from 10 mg L^−^^1^ to 40 mg L^−^^1^, the removal efficiencies after 2 h were 95.85%, 95.27%, 85.07%, and 83.99%, respectively (Figure 3d). Typically, higher pollutant concentrations lead to decreased degradation efficiency and slower reaction rates. While increasing the initial concentration of wastewater could theoretically reduce mass transfer limitations and improve degradation, the results indicate that SMX degradation is not primarily driven by electrochemical reactions. The decrease in efficiency at higher concentrations is likely due to a limited number of active sites and reactive substances on the particle electrode. As a result, the quantity of pollutants that can be adsorbed and degraded is fixed, making lower pollutant concentrations more favorable for effective degradation [46]. Additionally, the accumulation of intermediate products may block active sites, hindering further adsorption and the continuation of the degradation progress. SMX contains refractory functional groups, such as the benzene rings, which further inhibit degradation as concentration increases [47]. Given that SMX is a recalcitrant organic pollutant typically found in water at levels ranging from ng L^−1^ to μg L^−1^, this study demonstrates that a three-dimensional particle electrode system is highly effective for removing low concentrations of refractory organic pollutants from water.

Figure 3e illustrates that, under constant test conditions, the degradation efficiencies for SMX were 64.39%, 68.11%, 72.66%, 92.75%, 95.27%, 99.45%, and 98.34% for particle electrode dosages of 0 g L^−^^1^, 0.02 g L^−^^1^, 0.05 g L^−^^1^, 0.1 g L^−^^1^, 0.2 g L^−^^1^, 0.3 g L^−^^1^, and 0.4 g L^−^^1^, respectively. As the dosage increased from 0 g L^−^^1^ to 0.3 g L^−1^, the degradation efficiency improved significantly. This is primarily due to the higher number of particle electrodes in the system, which provided more reactive sites, thereby improving the adsorption of SMX onto the electrodes [48]. Consequently, the adsorbed SMX could efficiently participate in redox processes. Additionally, the higher particle dosage led to an increased concentration of Fe^2+^ and Ni^2+^ in the solution, which favored their complete reaction with H_2_O_2_ generated at the cathode, producing more reactive species such as •OH [49]. However, beyond 0.3 g L^−1^, the degradation effectiveness slightly decreased. This is likely due to the excessive number of particle electrodes causing accumulation and congestion, which increased the short-circuit current and introduced fluctuations in the reaction current, thereby diminishing the system’s performance [49]. Moreover, an excess of catalysts can lead to resource wastage and reduced reaction rates due to the polymerization effects [50,51]. Based on these findings, the optimal dosage for the system was determined to be 0.2 g L^−1^.

Na_2_SO_4_, a non-reactive electrolyte, is commonly utilized to improve the conductivity of wastewate [5]. As the Na_2_SO_4_ concentration increased from 0 to 0.10 mol L^−1^, the SMX degradation efficiency rose significantly from 7.21% to 98.79%, underscoring the critical role of conductivity in the electrocatalytic degradation process. However, further increasing the Na_2_SO_4_ concentration to 0.40 mol L^−1^ resulted in only a marginal improvement in degradation efficiency, indicating that beyond a certain threshold, additional electrolyte offers diminishing returns.

### 2.4. Stability of the EC/FeNi5@BC9 System

Evaluating the performance of the EC/FeNi5@BC9 system involved assessing both its structural stability and overall applicability [52]. The results show that the FeNi5@BC9 catalyst maintained an SMX removal efficiency exceeding 80% after five cycles, with removal efficiencies of 98.64%, 93.30%, 91.02%, 88.12%, and 81.07% for each cycle (Figure 4a). The ICP-OES analysis revealed that the leaching of Fe and Ni ions remained below 0.25 mg L^−^^1^ after each use, indicating negligible metal loss (Figure 4b). These findings suggest that the FeNi5@BC9 catalyst exhibits excellent stability, reusability, and environmental compatibility for practical applications.

### 2.5. Evaluation of the EC/FeNi5@BC9 System Applicability

#### 2.5.1. Degradation Performance of Different Organics

To assess the sustained effectiveness of the EC/FeNi5@BC9 system in treating hard-to-degrade organic pollutants, degradation tests were conducted using model substrates such as rhodamine B, bisphenol A, tetracycline, levofloxacin, and 2,4-dichlorophenol. As shown in Figure 4c, the system effectively degraded pollutants at an initial concentration of 20 mg L^−^^1^, achieving removal efficiencies of 94.97%, 98.8%, 80.68%, 72.10%, and 96.3%, respectively. These results confirm the robust and versatile capability of the EC/FeNi5@BC9 catalytic for degrading a variety of organic pollutants, including antibiotics. Increasing the catalyst dosage could further enhance the completeness and efficiency of pollutant degradation within this system.

#### 2.5.2. Effects of Different Environmental Conditions

In aqueous systems, ROS can be disrupted by the presence of inorganic anions such as Cl^−^, CO_3_^2−^, SO_4_^2−^, NO_3_^−^, and HCO_3_^−^ during the degradation process. It was found that Cl^−^ had a minimal impact on SMX degradation (Figure 5a). However, the degradation of SMX was notably inhibited by CO_3_^2−^ and HCO_3_^−^ (Figure 5b,e), likely due to their ability to increase the system’s pH and react with **•**OH, SO_4_^−^**•**, **•**O_2_^−^, and ^1^O_2_ [31]. The addition of SO_4_^2−^ and NO_3_^−^ to the EC/FeNi5@BC9 system initially decreased the degradation efficiency, but it subsequently recovered (Figure 6c,d). Overall, the EC/FeNi5@BC9 system demonstrated strong resilience to interference from inorganic anions, including Cl^−^, CO_3_^2−^, SO_4_^2−^, NO_3_^−^, and HCO_3_^−^.

In order to better simulate real-world conditions, SMX wastewater was prepared using Shenzhen Bay seawater and tap water. The pH levels of SMX wastewater samples, whether made with Shenzhen Bay seawater, tap water, or laboratory-deionized water, were comparable, ranging from 6.0 to 6.1. While both deionized water and tap water exhibited minimal total organic carbon (TOC) levels, Shenzhen Bay seawater contained a TOC of 4.46 mg L^−1^. The results demonstrated that SMX wastewater prepared with either Shenzhen Bay seawater or tap water achieved effective degradation performance (Figure 5f), underscoring the EC/FeNi5@BC9 system’s considerable potential for treating organic wastewater under real-world conditions.

### 2.6. The SMX Degradation Mechanism of the EC/FeNi5@BC9 System

#### 2.6.1. Evaluation of the ROS

Free radical scavenging experiments were conducted using TBA, MeOH, p-BQ, and L-his as scavengers for •OH, SO_4_^−^•, •O_2_^−^, and ^1^O_2_, respectively. As shown in Figure 6a, the addition of p-BQ to the EC/FeNi5@BC9 system resulted in a sharp decrease in degradation efficiency, from 99.49% to 16.56%, emphasizing the predominant role of •O_2_^−^ in the degradation process which contributed 71% to the overall efficiency [53]. With the addition of TBA, MeOH, and L-his, the degradation efficiency dropped to 79.48%, 81.82%, and 88.01%, respectively. These results indicate that at an SO_4_^2−^ concentration of 0.1 M, •OH, SO_4_^−^•, and ^1^O_2_ were minor reactive species, contributing only 17%, 2%, and 10%, respectively.

The active species’ contribution rates within the EC/FeNi5@BC9 system under varying SO_4_^2−^ concentrations were studied (Figure 6b,c). At a SO_4_^2−^ concentration of 0.025 M, the contributions of **•**O_2_^−^ and SO_4_^−^**•** were nearly negligible, whereas ^1^O_2_ and **•**OH contributed 56.0% and 34.0%, respectively, to the total. This is likely due to the limited availability of SO_4_^2^ electrolytes and the lack of a strong external circuit [54]. With an increase in SO_4_^2−^ concentration from 0.05 M to 0.10 M, the contribution of •O_2_^−^ rose significantly, aligning with the understanding that higher SO_4_^2−^ concentrations improve solution conductivity and electron efficiency [55]. However, beyond 0.10 M, the contribution of **•**O_2_^−^ initially increased but then decreased, suggesting that excessive SO_4_^2−^ can quench **•**O_2_^−^. When SO_4_^2−^ concentrations exceed 0.10 M, the resulting highly polar environment impedes the movement and dispersion of other species toward the catalyst’s active site. Moreover, an excess of SO_4_^2−^ can quench other reactive species or generate less potent oxidants, suppressing the redox reaction [56]. In conclusion, the superior SMX removal efficiency in the EC/FeNi5@BC9 system is due to the synergistic action of multiple active species (•O_2_^−^ > •OH > ^1^O_2_ > SO_4_^−^•).

To further validate the existence of the aforementioned active species, this study employed DMPO for the detection of free radicals (•O_2_^−^, •OH, SO_4_−•) and utilized TEMP to identify singlet oxygen (Figure 6d). The active species were then characterized through electron EPR testing [53]. The identification of •OH and SO_4_^−^• within the EC/FeNi@BC system was achieved by introducing DMPO, which successfully captured these radicals, resulting in a pronounced radical characteristic peak with a 1:2:2:1 ratio [57]. Similarly, in the EC/FeNi@BC system utilizing methanol as a solvent and DMPO for the detection of superoxide radicals, the EPR characterization revealed a distinct sixfold peak, which validated that •O_2_^−^ was indeed present in the system [58]. These findings align with the outcomes observed in previous quenching experiments. This body of evidence strengthens the assertion that the EC/FeNi@BC system facilitated the degradation of SMX through a synergistic interaction between multiple free radicals. The involvement of multiple radical types improves the system’s effectiveness and underscores its complex degradation mechanism.

Based on the experimental findings, a mechanistic pathway for the oxidative degradation of SMX in the EC/FeNi@BC three-dimensional particle electrode system is proposed. SMX degradation proceeds via reactions occurring both on the electrode surface and within the solution. Dissolved oxygen, adsorbed on the cathodic surface of the FeNi@BC particle electrode with high specific surface area, undergoes a single-electron reduction, generating the highly electrophilic radical •O_2_^−^. This radical preferentially targets the electron-rich sites on SMX, initiating its degradation. Concurrently, part of the •O_2_^−^ species undergoes an additional single-electron reduction, leading to the formation of H_2_O_2_. Subsequently, H_2_O_2_ reacts with the cathode surface catalyst, producing •OH radicals, which further oxidize SMX molecules and partially degraded intermediates. The regeneration of the catalyst by electrons on the cathode surface enhances the system’s efficiency for SMX degradation. In parallel, a portion of the catalyst dissolved in the solution reacts with H_2_O_2_ to yield •OH radicals, facilitating SMX oxidation and removal through a homogeneous-phase electro-Fenton-like process.

#### 2.6.2. Performance Evaluation

To better understand the activation mechanism within the EC/FeNi@BC system, the FeNi5@BC9 catalyst was recovered and subjected to post-reaction characterization. As illustrated in Figure 7a, the surface morphology of FeNi5@BC9 remained largely unchanged after the reaction, indicating that the catalyst retained its structural integrity throughout the process. XRD analysis of the fresh and used FeNi5@BC9 further confirmed the stability of the catalyst’s framework (Figure 7b). However, a slight attenuation in the diffraction peaks corresponding to Fe (Iron, delta) and NiO was observed post-reaction, suggesting partial oxidation of Fe and NiO to Fe^3+^ and Ni^2+^, respectively, during the electrocatalytic oxidation process. The composites before and after the reaction were analyzed via XPS (Figure 7c–f). The FeNi5@BC9 composite exhibited characteristic peaks corresponding to two main sets of binding energies: 712.32 eV (Fe^2^⁺), 722.66 eV (Fe^3^⁺), 873.36 eV (Ni^2^⁺), and 856.25 eV (Ni^3^⁺). Before the reaction, the Fe^2^⁺ and Fe^3^⁺ contents were 34.58% and 18.03%, respectively, while the Ni^2^⁺ and Ni^3^⁺ contents were 24.02% and 23.74%, respectively. After the reaction, the Fe^2^⁺ content increased to 36.58%, and the Ni^3^⁺ content increased slightly to 24.00%. Additionally, shifts in the Fe^3^⁺ peaks toward lower binding energies and the Ni^2^⁺ peaks toward higher binding energies were observed, indicating the presence of an electron transfer pathway within the material. This electron transfer facilitates the reduction of Fe^3^⁺ to Fe^2^⁺ and the oxidation of Ni^2^⁺ to Ni^3^⁺. These findings suggest an interconversion between Fe^3^⁺/Ni^2^⁺ and Fe^2^⁺/Ni^3^⁺ on the surface of FeNi5@BC9 particles, with the associated electron transfer significantly enhancing the degradation of pollutants [59]. Furthermore, the O 1s spectra revealed a reduction in the relative content of adsorbed oxygen (Oads) during the catalytic reaction, from 73.16% before the reaction to 56.71% after (Figure 7f). This decrease indicates the active participation of Oads in the efficient degradation of pollutants during electrocatalysis.

#### 2.6.3. SMX Degradation Pathway

To investigate the mechanism underlying SMX degradation, LC-MS analysis was employed to identify and study the intermediates generated during the process. As illustrated in Figure 8, SMX degradation follows three primary pathways, each leading to the formation of different small molecules: the loss of aniline (*α*), the cleavage of the S-N bond (*β*), and the removal of the isoxazole ring (*γ*).

Pathway A: Degradation begins with the hydroxylation of the C=C bond in SMX, resulting in the formation of the initial product P1-1 due to the addition of hydroxyl groups to the isoxazole ring [60]. The carbon atoms adjacent to the amino group on the aniline ring are highly reactive, leading to further hydroxylation and the formation of P1-2 [61]. Continued hydroxylation and subsequent reactions break the S-N and S-C bonds, ultimately yielding P1-3 and P1-4 [62].

Pathway B: This pathway primarily involves the cleavage of the isoxazole ring. Free radicals attack the ring, producing the product P2-1 [63]. The most reactive site for these attacks is the nitrogen position on the aniline ring, resulting in the replacement of the amino group with a hydroxyl group, forming P2-2 [64]. Further cleavage of the S-N bond generates P2-3 [65]. Research indicates that the S-N bond is particularly susceptible to attack due to its electron-rich nature [66].

Pathway C: This pathway focuses on the cleavage of the SMX molecule through hydroxyl group attacks, producing P3-1, which subsequently converts to benzenesulfinic acid and eventually to phenylthiophenol [67]. The oxazole ring, being less reactive than the aniline ring, results in the formation of P4-1 after the S-N bond breaks [68]. This intermediate product is further oxidized to P4-2 [69]. Ultimately, SMX degrades into small organic molecules and undergoes mineralization. Furthermore, TOC analysis revealed a removal efficiency of 85.19% after 120 min (Appendix A), indicating that SMX was effectively mineralized into CO_2_ and H_2_O [70,71]. All degradation products are detailed in Appendix A.

## 3. Materials and Methods

### 3.1. Chemical Reagents

Sodium chloride (NaCl), sodium carbonate (Na_2_CO_3_), sodium bicarbonate (NaHCO_3_), sodium phosphate (Na_3_PO_4_), sodium nitrate (NaNO_3_), sodium sulfate (Na_2_SO_4_), sulfuric acid (H_2_SO_4_), sodium hydroxide (NaOH), ferrous chloride tetrahydrate (FeCl_2_·4H_2_O), and nickel chloride hexahydrate (NiCl_2_·6H_2_O) were sourced from Sinopharm Chemical Reagent Co., Ltd., Shanghai, China. Methanol (MeOH), tert-butyl alcohol (TBA), L-Histidine (L-His), 1,4-Benzoquinone (p-BQ), and sulfamethoxazole (SMX) were obtained from national pharmaceutical reagent, Shanghai China. Macklin Biochemical Technology Co., Ltd., Shanghai, China supplied 5-dimethyl-1-pyrroline N-oxide (DMPO, 97%) and 2,2,6,6-tetramethyl-4-piperidinol (TEMP, 99%). Additionally, Rhodamine B, bisphenol A (BPA), tetracycline, levofloxacin (LEV), and 2,4-dichlorophenol (2,4-DCP) were provided by Sinopharm Chemical Reagents Co., Ltd., Shanghai, China.

### 3.2. Preparation of the Particle Electrode

Previous studies have shown that the wetland plant Iris sibirica L. effectively removes heavy metals due to its high accumulation capacity and substantial biomass production [13]. Detailed methods for its cultivation and the preparation of FeNi@BC and FeNi-BC catalysts are provided in the Appendix A.

Research shows that metal ions tend to accumulate more in the roots than in the stems [14]. Therefore, roots of Iris sibirica L. subjected to varying concentrations of Fe and Ni stress (0.00, 300.00, and 500.00 mg L^−1^) were used as raw materials. These were then pyrolyzed at temperatures of 500 °C, 700 °C, and 900 °C, respectively. The biochar was named based on different metal ion stress concentrations and pyrolysis temperatures. For example, biochar produced from biomass exposed to a metal ion concentration of 500 mg L^−1^ and pyrolyzed at 900 °C was designated as FeNi5@BC9.

Fe and Ni were deposited onto the surface of BC9 using a pyrolytic impregnation method, and the resulting samples were labeled as synthetic FeNi-BC catalysts, with the ion concentrations of Fe and Ni matching those in FeNi5@BC9. In this study, FeNi5@BC9 was selected from all bio-inspired FeNi@BC catalysts for its superior catalytic performance (Appendix A), while BC9 and FeNi-BC were used as control samples.

### 3.3. Analytical Methodologies

The surface morphology and elemental composition of the particle electrode were analyzed using a scanning electron microscope (SEM, FEI ETEM 80-300, Rock Hill, SC, USA) integrated with energy-dispersive spectroscopy (EDS). Specific surface area (SSA) and pore size distribution of the materials were determined by a surface area and pore size analyser (Micromeritics Tristar II) from Micromeritics (Shanghai, China) Instruments Co. For analyzing the crystal structure and the metal’s valence states on the particle electrode’s surface, X-ray diffraction (XRD, D8 ADVANCE, Karlsruhe, Germany) and X-ray photoelectron spectroscopy (XPS, Thermo Fisher, Waltham, MA, USA, ESCALAB 250Xi) were employed. Electrocatalytic activity and repolarization capabilities were assessed with an electrochemical workstation (EW, CHI 760E, Shanghai, China). The removal efficiency of SMX was determined using high-performance liquid chromatography (HPLC, Arc HPLC-2998). The degradation mechanism was investigated with triple quadrupole liquid chromatography–mass spectrometry (LC-MS, TRIPLE TOF 4600, Agilent, Santa Clara, CA, USA).

### 3.4. SMX Degradation Process

Figure 9 illustrates the reactor used for the three-dimensional catalytic oxidation system. In the experiment, 10 mg of FeNi5@BC9 and 50 mL of SMX solution (20 mg L^−1^) were added to a polypropylene electrochemical reaction tank, where the mixture was electrolyzed for 120 min under controlled conditions (voltage = 8 V, pH = 3, and Na_2_SO_4_ dose = 0.10 M). Samples were collected at predetermined intervals and filtered through a 0.22 μm polyethersulfone membrane. The efficiency of SMX removal was assessed using a HPLC system equipped with a C18 reversed-phase column (4.6 × 250 mm) and a UV detector set to 269 nm. The mobile phase consisted of acetonitrile and 0.1% formic acid in a 40:60 (*v*1/*v*2) ratio, with a flow rate of 0.8 mL min^−1^ and an injection volume of 20 μL.

The degradation (%) of SMX was estimated based on the following Equation (8):SMX degradation rate (%) = (C_0_ − C_t_)/C_0_,(8)

The initial concentration of SMX is expressed as C_0_ (mg L^−1^), and the concentration of SMX at each time during the degradation process is represented by C_t_ (mg L^−1^).

## 4. Conclusions

In the EC degradation process, challenges such as poor mass transfer, suboptimal mineralization rates, and limited current efficiency have hindered broader applications. To address these issues, this study synthesized spherical particle electrodes (FeNi@BC) with enhanced electrocatalytic performance using a bio-inspired preparation method. These electrodes were integrated into an EC reaction tank to create a three-dimensional catalytic oxidation system, referred to as EC/FeNi@BC. This novel system significantly improved the degradation rate of SMX, achieving 0.0456 min^−1^, compared to the EC (0.0084 min^−1^) and EC/FeNi-BC (0.0121 min^−1^) systems. Various operational parameters, including voltage, pH, initial SMX concentration, particle electrode dosages, and Na_2_SO_4_ concentration, were systematically investigated for their effects on SMX degradation. Free radical scavenging and EPR experiments revealed that the superior removal efficiency of SMX in the EC/FeNi@BC system was attributed to the synergistic action of multiple reactive species (•O_2_^−^, •OH, ^1^O_2_, and SO_4_^−^•). XRD and XPS analyses confirmed that the primary active sites of the FeNi@BC electrode are the metal oxides of Fe and Ni. Measurements from the electrochemical workstation further revealed that FeNi@BC demonstrated strong electrocatalytic activity and repolarization capability. Notably, after five cycles, the EC/FeNi@BC system maintained a high SMX removal efficiency of 81.07%. Additionally, LC-MS analysis identified 12 intermediates generated during the degradation process. This study provides new insights into advanced water treatment technologies.

## Figures and Tables

**Figure 1 ijms-25-13579-f001:**
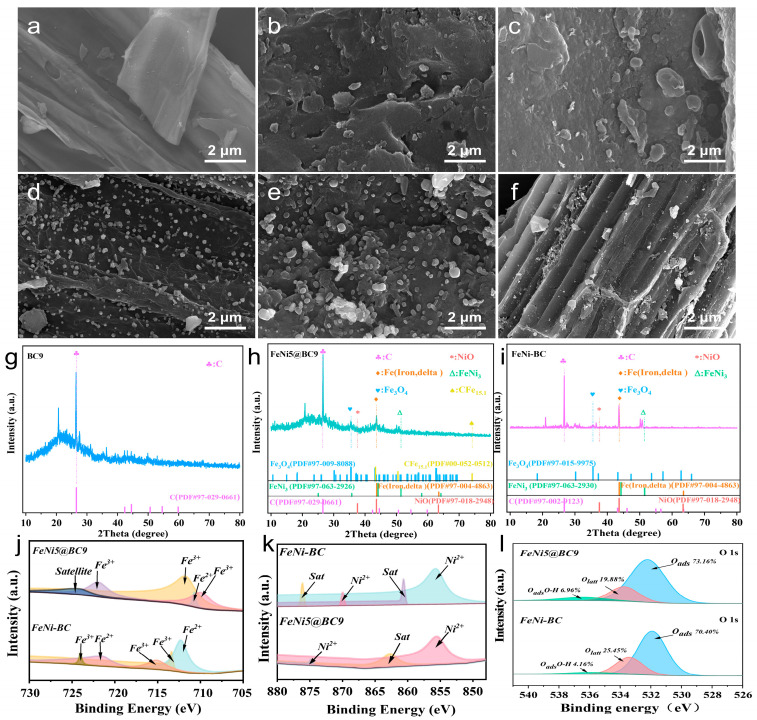
SEM images of BC (**a**), FeNi5@BC5 (**b**), FeNi5@BC7 (**c**), FeNi5@BC9 (**d**,**e**) and FeNi-BC (**f**); the XRD patterns of BC (**g**), FeNi5@BC9 (**h**) and FeNi-BC (**i**); XPS spectra of Fe 2p (**j**), Ni 2p (**k**) and O 1s (**l**).

**Figure 2 ijms-25-13579-f002:**
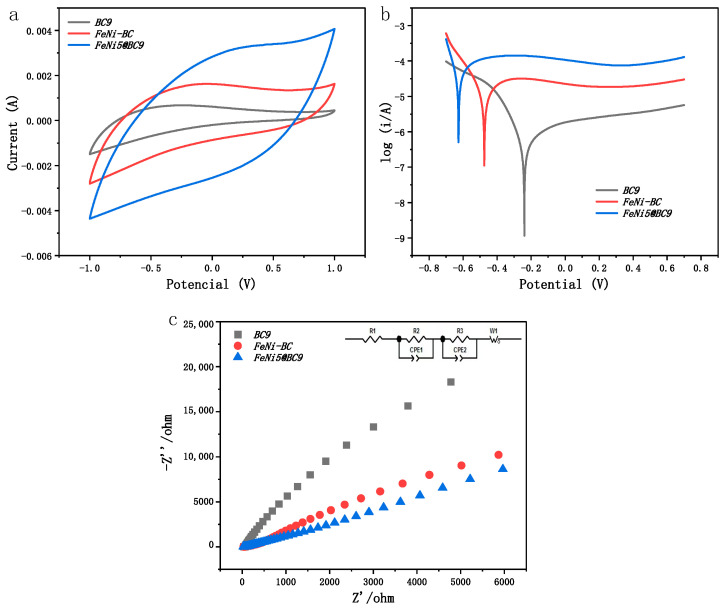
Cyclic voltammetry curve (**a**), Tafel polarization curve (**b**) and electrochemical impedance spectroscopy (**c**) of BC, FeNi-BC, and FeNi5@BC9.

**Figure 3 ijms-25-13579-f003:**
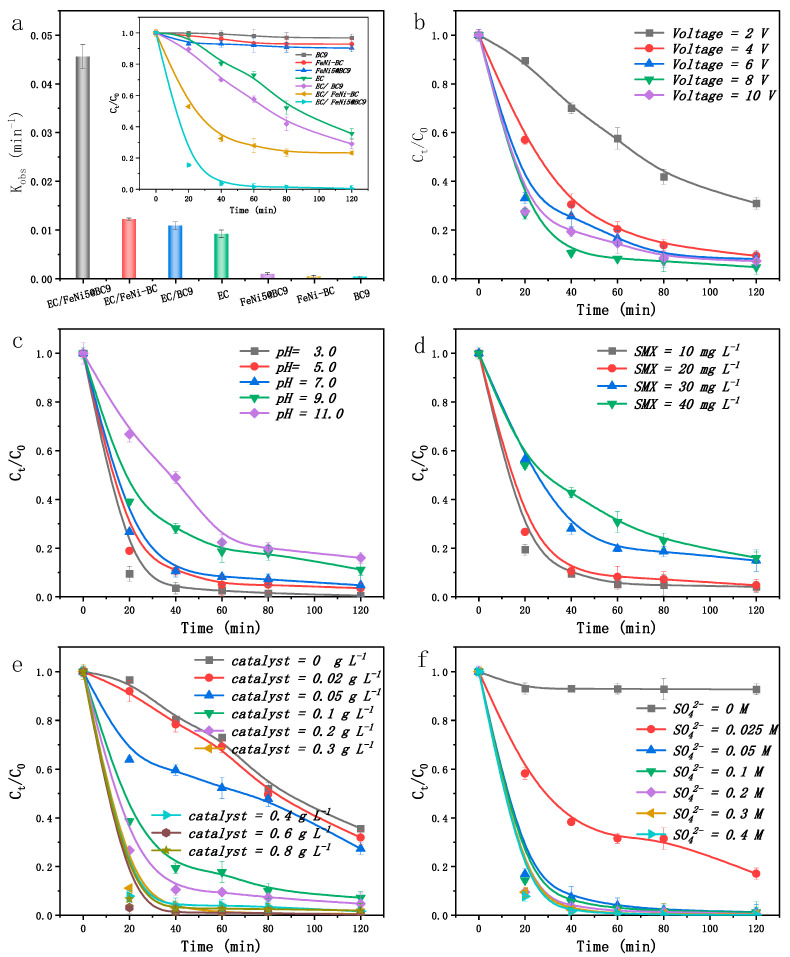
Removal curves of SMX in the different degradation systems and rate constants of the pseudo-first-order kinetic model (**a**), the effect of voltage (**b**), pH (**c**), SMX initial concentration (**d**), catalyst dosage (**e**), and Na_2_SO_4_ dose (**f**) on the degradation of SMX (Voltage = 8 V, pH = 3, [SMX]_0_ = 20 mg L^−1^, the dose of catalyst = 10 mg, Na_2_SO_4_ dose = 0.10 M, and the degradation time = 120 min).

**Figure 4 ijms-25-13579-f004:**
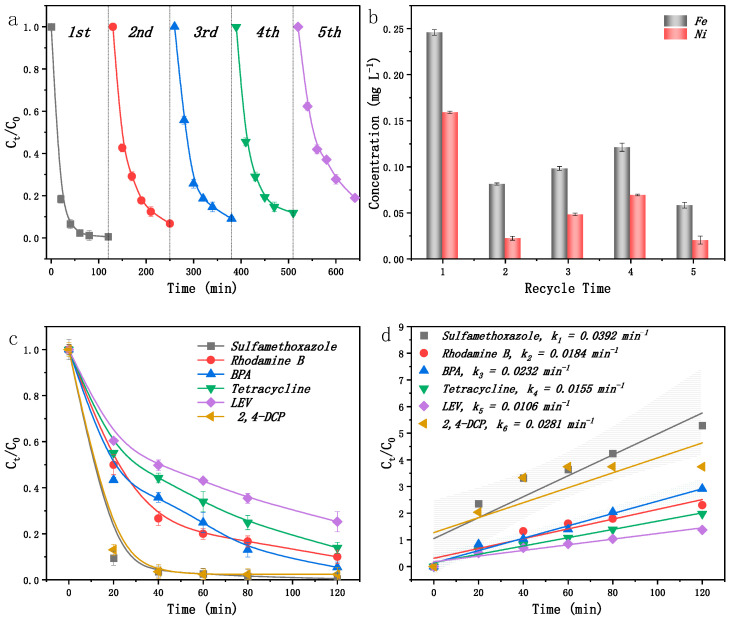
Structural stability of five cycles of SMX degradation experiment (**a**), five cycles of Fe, Ni ion leaching (**b**), the degradation and Kobs of different pollutants (**c**,**d**).

**Figure 5 ijms-25-13579-f005:**
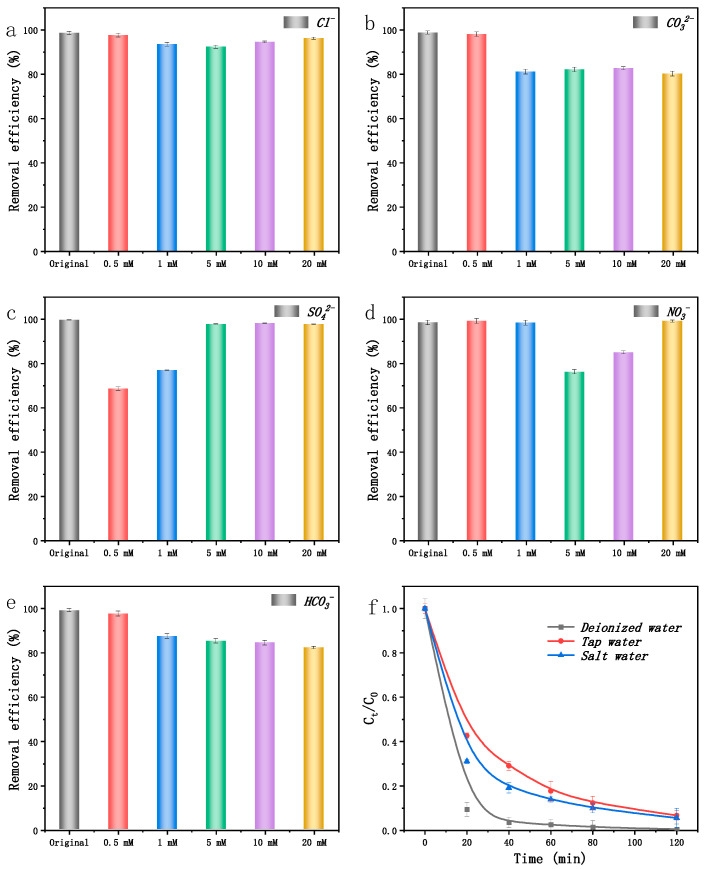
Effect of Cl^−^ (**a**), SO_4_^2−^ (**b**), NO_3_^−^ (**c**), CO_3_^2−^ (**d**), HCO_3_^−^ (**e**), and solvents (**f**) on SMX degradation. (Voltage = 8 V, pH = 3, the dose of catalyst = 10 mg, [SMX]_0_ = 20 mg L^−1^, Na_2_SO_4_ dose = 0.10 M, the degradation time = 120 min).

**Figure 6 ijms-25-13579-f006:**
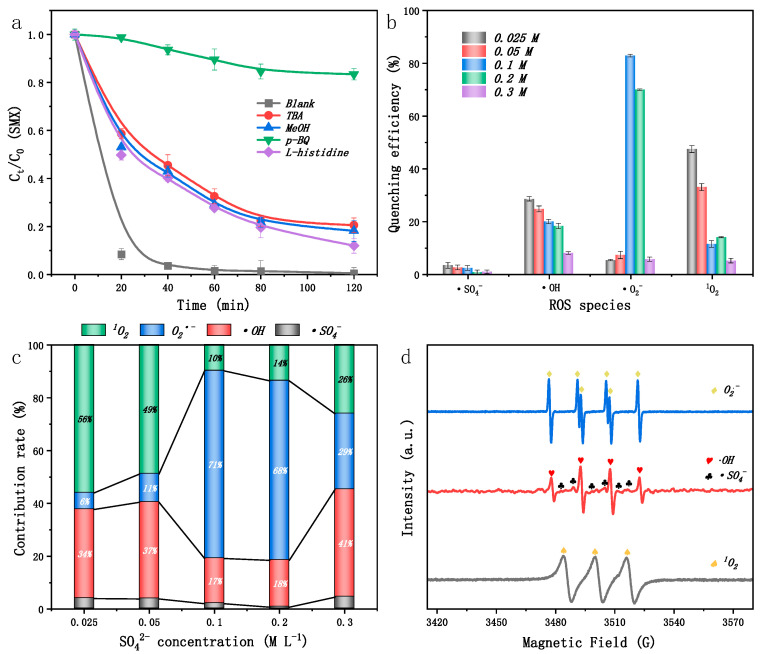
The degradation efficiency of SMX under different quenching conditions (**a**), ROS species individual contribution (**b**) and synergistic contribution (**c**), EPR spectra of •O_2_^−^, •OH, ^1^O_2_, and SO_4_^−^• in EC/FeNi@BC systems (**d**). (Voltage = 8 V, pH = 3, [SMX]_0_ = 20 mg L^−1^, the dose of catalyst = 10 mg, Na_2_SO_4_ dose = 0.10 M, and the degradation time = 120 min).

**Figure 7 ijms-25-13579-f007:**
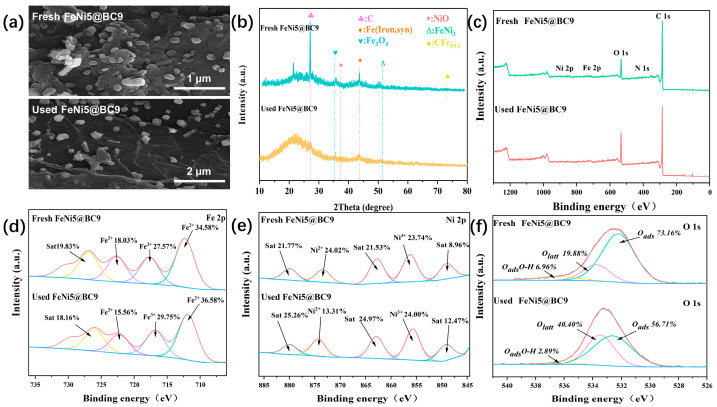
SEM image (**a**), XRD pattern (**b**) and XPS full spectra (**c**), Fe 2p (**d**), Ni 2p (**e**), and O 1s (**f**) of FeNi5@BC9 before and after reaction.

**Figure 8 ijms-25-13579-f008:**
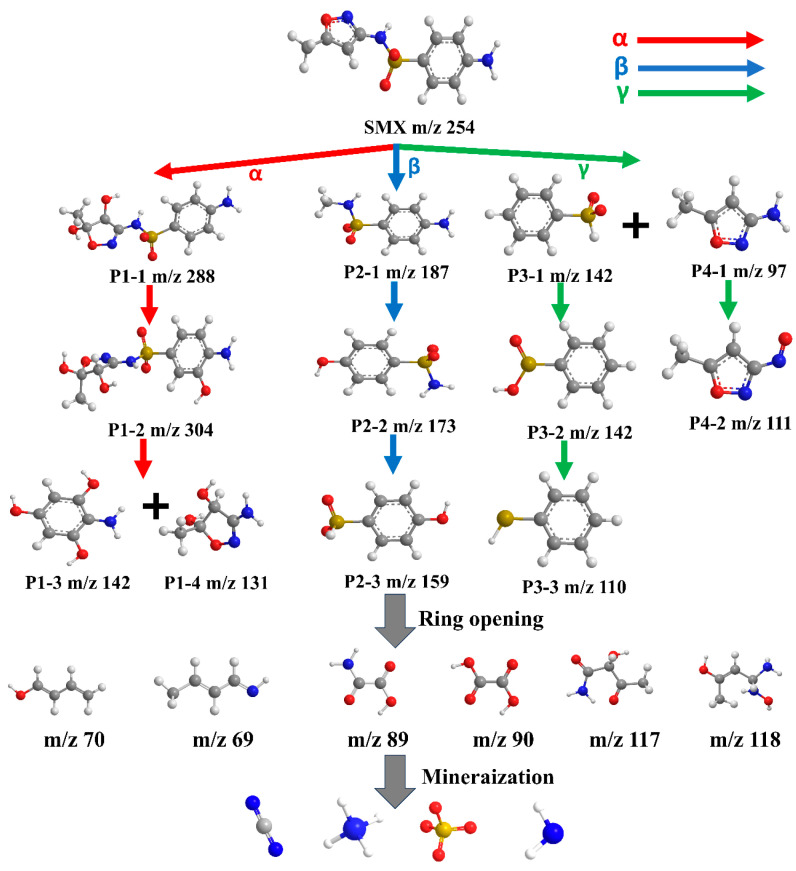
The degradation pathway of SMX.

**Figure 9 ijms-25-13579-f009:**
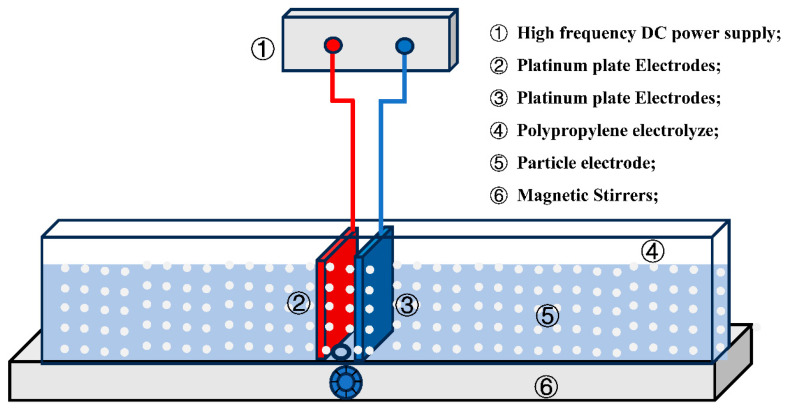
The experimental device.

## Data Availability

Data are contained within the article and Appendix A.

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
