# Peer review of "Electrochemical Degradation of Sulfamethoxazole Enhanced by Bio-Inspired Iron-Nickel Encapsulated Biochar Particle Electrode"

_ijms, 2024, doi:10.3390/ijms252413579_

Round 1
Reviewer 1 Report
Comments and Suggestions for Authors
The authors have thoroughly discussed the electrochemical degradation of sulfamethoxazole using a bio-inspired iron-nickel encapsulated biochar electrode.
The overall quality of the work is commendable.
The authors must address the following shortcomings before publication:
1. Provide EDS spectra.
2. Repeat the EIS analysis and present appropriate circuit models.
3. Conduct a comprehensive English language revision.
4. Fix typographical errors.
Comments on the Quality of English LanguageNeed to improve
Reviewer 2 Report
Comments and Suggestions for Authors
In this work, the authors prepared EC/FeNi@BC electrode for efficient degradation of sulfamethoxazole, reaching 0.0456 min-1. Some characterizations were used to demonstrate the performance origin and catalysis mechanism. However, some important issues should be solved before acceptance.
1. In Figure 2g, XRD refinement should be performed to exactly extract the detailed content of each phase in the composite. Please refer to the paper with DOI of 10.1016/j.jechem.2023.03.033.
2. Oxygen species are important for the redox reactions. O 1s XPS spectrum should be measured and analyzed to demonstrate this contribution. Please refer to the paper with DOI of 10.1038/s41467-020-19433-1.
3. Necessary structural and electronic structural characterizations should be conducted on samples after reactions.
4. The analysis of EIS is too simple. Please add more discussions.
Comments on the Quality of English LanguageThe English could be improved to more clearly express the research.
